# Hybrid Cheeses—Supplementation of Cheese with Plant-Based Ingredients for a Tasty, Nutritious and Sustainable Food Transition

Blandine M. L. Genet [1],*, Guillermo Eduardo Sedó Molina [1], Anders Peter Wätjen [1], Giovanni Barone [2], Kristian Albersten [3], Lilia M. Ahrné [2],†, Egon Bech Hansen [4],† and Claus H. Bang-Berthelsen [1],*,†

[1] Research Group for Microbial Biotechnology and Biorefining, National Food Institute, Technical University Denmark, Kemitorvet, Building 202, 2800 Kongens Lyngby, Denmark
[2] Department of Food Science, University of Copenhagen, Rolighedsvej 30, 1958 Frederiksberg, Denmark; lilia@food.ku.dk (L.M.A.)
[3] Thise Dairy, Sundsørevej 62, Thise, 7870 Roslev, Denmark
[4] Research Group for Gut, Microbes and Health, National Food Institute, Technical University Denmark, Kemitorvet, Building 202, 2800 Kongens Lyngby, Denmark
*  Correspondence: blagen@food.dtu.dk (B.M.L.G.); claban@food.dtu.dk (C.H.B.-B.)
†  Last author shared.

**Abstract:** With increasing awareness of the impact of food on the climate, consumers are gradually changing their dietary habits towards a more plant-based diet. While acceptable products have been developed in meat analogues and non-fermented dairy products, alternative fermented dairy products such as yogurt and particularly ripened hard and semi-soft cheese products are not yet satisfactory. Since the cheese category has such a broad range of flavors and applications, it has proven complicated to find plant-based sources able to mimic them in terms of texture, meltability, ripening and flavor. Moreover, plant-based dairy alternatives do not provide the same nutritional supply. New technological approaches are needed to make cheese production more sustainable, which should be integrated in the already existing conventional cheese production to ensure a fast and cost-efficient transition. This can be tackled by incorporating plant-based components into the milk matrix, creating so-called "hybrid cheeses". This review will discuss the challenges of both animal- and plant-based cheese products and highlight how the combination of both matrices can associate the best properties of these two worlds in a hybrid product, reviewing current knowledge and development on the matter. Emphasis will be drawn to the selection and pre-processing of raw materials. Furthermore, the key challenges of removing the off-flavors and creating a desirable cheese flavor through fermentation will be discussed.

**Keywords:** cheese; plant based; sustainability; hybrid cheese; fermentation; lactic acid bacteria; alternative dairy

## 1. Introduction

The production of food from animal sources has a significant impact on the climate and on the destruction of ecosystems. Dairy, and particularly cheese, is not an exception. Indeed, when ranking food categories for their greenhouse gas (GHG) emissions, cheese comes fifth, after beef herds, lamb and mutton, beef from dairy herds and crustaceans, and before pig, fish, poultry, eggs and milk [1]. Even though the dairy sector has become more efficient in the past 50 years [2], there is a limit to how sustainable animal farming can become; therefore, drastic changes towards more plants for food are needed in the food system to maintain global warming below 2 °C [3]. Moreover, the Eat-Lancet commission highlighted in their report the urgency to reduce the consumption of animal-based food and to increase the proportion of plant-based items in the diet, both for health and sustainability reasons [4]. Consumers are increasingly concerned about sustainability and the actions they

can undertake to reduce their impact. Changing diet towards a more plant-based one is an obvious choice that many are undertaking. For example, in Denmark, 3% of the population declared themselves vegetarians in 2022, in comparison to 1.8% in 2017. In the age range of 18 to 34 years old, the numbers rise to 7.4% [5]. This can be seen in the development of new plant-based analogues, such as meat substitutes [6,7], milk analogues [8], and yogurt [9]. Other plant-based products are in the early stages of development, such as alternatives to eggs, ice cream or fish [10]. The main drivers for consumers to change to a more plant-based diet are mainly health concerns for flexitarians and ethical reasons for vegans and vegetarians [11]. However, changing habits is difficult and dietary habits are not an exception. For new and more sustainable products to be adopted as a regular part of the diet, the product must be pleasant to consume; otherwise, the curiosity purchase will not transform into a regular purchase. Plant-based alternatives (PBA) are mainly developed with the aim of replicating the properties of existing animal products. Due to the different functionalities of the used materials, the perfect match is difficult to reach, which generally results in low acceptance [10]. Plant-based cheese alternatives (PBCA) are massively arriving on the market, with solutions for many different types of cheeses such as cream cheese, brine cheese (mozzarella, feta), blue cheese or longer-ripened cheeses such as edam or gouda. They are mostly made with coconut oil, starches, nuts or soy [12]. However, the products currently available have not yet reached satisfying quality in several aspects such as sensory, nutrition and sustainability, depending on the base used, which will be reviewed in the next paragraphs.

First, the sensory properties do not live up to the expectations of consumers. A major sensory issue with soy-based cheese alternatives (CA) is the beany flavor and grittiness of the mouthfeel, which could be alleviated via fermentation, processing or blending different plant matrices together [13]. Falkeisen et al. [14] measured sensory aspects and emotional response to five different plant-based (PB) shredded-cheddar-type products. They found that even though some products performed better than others, they were overall not liked, and even participants regularly consuming PBCA assigned it low liking scores. Also, they reported the conflict of participants wanting to consume PBCA but not doing so because the taste was not satisfactory [14]. Giacalone et al. provided a detailed review of the different sensory and consumer behavior aspects leading to the different acceptance levels of consumers towards PBA [15]. One of the highlights is that mimicking the exact properties of conventional products is counterproductive with increasing familiarity to plant-based alternatives: the more familiar consumers are with alternatives, the less they want them to resemble real animal products [16]. This is an opportunity to create products with new tastes and to embrace the plant-based aspect. However, overall liking of the product is the crucial criterion for omnivores and flexitarians to consume PBA. In contrast to vegetarians and vegans, they will not compromise on taste and might have less information about the environmental impact of different food products [17]. The greatest impact can be reached by convincing most consumers to switch to a more plant-based diet, and therefore the focus when developing a new sustainable product should be the sensory aspects [16].

Secondly, changing the matrix from animal milk to plant-based dairy alternatives (PBDA) does not provide the same nutritional properties. This can lead to severe deficiencies of which consumers are not always aware. Particularly concerned are young children, elderly people and women in the reproducing stage of their life [18]. Dairy products, and especially cheese, are important sources of macronutrients (protein and fats) and micronutrients (calcium, vitamin B2, vitamin B12, vitamin D and iodine) [19,20]. Few studies about the nutritional value of plant-based cheeses and comparison with animal counterparts have been conducted so far [18,21–23], while efforts were mostly concentrated on alternative milk products. These point out that the main nutritional challenges for PBCA are the lack of protein, calcium, vitamins (B2, B12 and D) and iodine and high amount of saturated fatty acids due to the use of coconut fat as a main ingredient [18,21–23]. Supplementation in micronutrients has proven to be an efficient way to achieve nutritional adequacy [18]. From the perspective of hybrid products, calcium can be fortified using the right propor-

tion of soluble and insoluble calcium salts to avoid effects on protein structure and salt precipitation [24]. Iodine could be fortified in the salting and brining steps, as has been shown by Wechsler et al. [25] in conventional cheese. On the macronutrient level, producing a nutritionally adequate product should be the primary concern when formulating PBCA, prioritizing protein-rich ingredients and not introducing high amounts of saturated fatty acids.

While the literature about the environmental impact of conventional dairy products is increasing in quantity, quality and accuracy, there are still very few studies on the impact of PBCA as such. A major barrier to creating these kind of data is the diversity of products, ingredients and processes involved, while traditional cheesemaking only requires milk from cows and, in a minority of cases, from other animals. In the review of 16 life cycle assessment (LCA) studies, Finnegan et al. [26] found that the overall environmental impact of cheese production is driven by the production of raw milk, accounting for 79 to 95% of the total global warming potential (GWP) of cheese, while the processing of raw milk into cheese only accounts for 2 to 18% of the GWP. Transportation and packaging appeared to be negligeable. Therefore, rethinking the source of raw materials is crucial to reduce the overall impact of cheese. Few studies have been conducted in the comparison between conventional dairy and PBDA, which was reviewed by Carlsson Kanyama et al. [27]. Overall, they found that PBDA had a much lower environmental impact than their conventional dairy counterparts. However, these types of studies are still in their infancy and many parameters are limiting, calling for the harmonization of the methods used, units used and information collected. Even if they seem clearly more sustainable, PBCA should also be questioned, depending on the choice of raw materials as a base and on their origin. Indeed, nuts have a high environmental and social impact, and even though the Eat-Lancet commission [4] recommends an increase in the consumption of nuts and seeds, they are precious ingredients and care should be taken to not lose their nutritional aspect during processing [28]. In their thorough assessment of the impact of nuts, Cap et al. [28] identified cashews and almonds as the worst-performing nuts. They are, however, very prominent as ingredients in PBCA. Moreover, the mostly manual post-harvest processing of cashew nuts has a disastrous impact on the health of the workers, who are mostly uneducated women whose voices are rarely heard in mainstream attention [29]. Therefore, to achieve a new cheese alternatives which can truly be considered sustainable, the ingredients for PBDA should be carefully selected in the perspectives of low environmental and social impact as well as local production. The use of industrial side streams is a promising source of material, allowing for both locality and circularity, which should be considered for hybrid cheese formulations, as has been investigated in yogurt alternatives with brewer's spent grain [30].

Animal cheese is an important source of nutrients and has a long history of traditional use, making it a category with wide diversity which is difficult to mimic and replace. However, the current consumption of cheese is not sustainable, and alternatives should be found. Even though numerous new PBCAs are brought to the market, their nutritional profile and sensory properties are not satisfactory. We hypothesize that creating a hybrid product combining the nutrition and functional properties of milk and milk proteins with the sustainability of plant-based components is a solution to lower the environmental impact of cheese consumption, while providing a high-quality product with both high nutritional profile and great flavor. In this literature review, we will investigate the current knowledge about mixing milk and plant-based proteins to create a gel network, summarize prior achievements in creating cheese analogue products by mixing both animal and plant-based milks as well as how can the raw ingredients be pre-processed to make them more functional. Finally, we will review how fermentation, which is a process commonly applied to cheese, is impacted by the change in matrix, and which opportunities it creates to improve the sensory and nutritional profile of products containing plant-based ingredients.

## 2. Recent Developments in Hybrid Cheeses

We define hybrid cheese as a cheese product made from milk and plant-based ingredients, where both components are retained in the final product to various concentrations. To the best of our knowledge, this terminology is not yet widely used, and only one publication was found using it [31]. Other terminologies are "mixed dairy and plant-based alternatives" or "dairy supplemented with plant-based ingredients".

Guyomarc'h et al. [32] reviewed the evolution of animal/plant mixed products, in the context of mixing milk, eggs and plant bases. They also summarize nutrition and digestibility aspects of such mixed foods. Mixing fast-digestible animal protein with slow-digestible plant protein is a good approach to fight against malnutrition and build up muscle mass [32]. Except for butter spreads, where parts of the fat are replaced with plant oils, hybrid products do not seem to have arrived on the European consumer markets yet. This might be mostly due to a lacking consumer base. However, to launch such products, it is important to evaluate consumer perception. Drigon et al. [33] observed the attitude towards mixed dairy products (milk, yogurt, solid dairy/cheese-like) in an online survey on French consumers. They found three types of consumers: the first are the ones who already consume dairy alternatives and have a negative opinion on milk (for health and environmental reasons), and have a relatively low opinion of such products because they already found their plant-based alternatives. The second type are consumers strongly attached to milk, who do not think that dairy is bad for the environment and even if, would not consider environment as a criterion to choose food. They have a bad perception of soy and are hard to convince if the taste is not spot on. The last category is made up of mostly women, self-declared omnivores, making altruistic and self-centered choices. They would change to mixes because they have fewer calories and a lower environmental impact, but the emotional properties would still be conserved because the product would be close to traditional use and familiarity [33]. There are, therefore, good incentives towards developing such hybrid products.

Milk-based cheese has two main components: protein and fat, with minimal amounts of carbohydrates and essential micronutrients. Therefore, plant-based replacements can be found for either the fat or the protein component, partially or fully, and in different proportions. A key challenge in hybrid cheese production is forming a protein network to obtain the desired structure, which will be discussed in the next section. Analogue cheese products appeared in the 1980s, and were developed to create cheap replacements for the ready-meal industry, such as pizza cheese toppings. Cost reduction was achieved by removing the milk fat, to be sold as butter, and replacing it with vegetable oils and fats. Such products permitted more flexibility in terms of functionality, were more consistent throughout seasons and had longer shelf-life, which made them easier to handle logistically [34]. Nowadays, consumers are more concerned about health aspects, and the replacement of dairy fat with plant fat can be an advantage. Furthermore, there is a general trend towards creating high-quality food products made for hedonic purposes to reduce environmental and health impacts.

Some research has been conducted in the direction of hybrid cheese, but it seems quite sporadic, and the methods used to assess the success of the formulation differ from study to study. Key factors for successful formulations are texture and sensory profiles. While the former can be conducted on machines with standardized protocols allowing for reproducibility, sensory analysis is not always conducted, and when applied, the robustness of the investigation is sometimes questionable. To significantly improve the quality of newly developed products, sensory analysis should receive more attention and be conducted in a more standardized way [13]. Table 1 summarizes the studies found to formulate hybrid cheeses by replacing either parts of the casein or fat with plant-based ingredients. It is notable that diverse types of cheese and plant-based replacements are being investigated.

Fermentation is traditionally applied in cheesemaking, as lactic acid bacteria (LAB) are essential for the acidification of the milk, enabling curd formation and increasing shelf life. Moreover, secondary cultures are also important for the development of specific

cheese aromas. The potential of fermentation for plant-based dairy alternatives will be discussed further down. However, while reports exist about fermented PBA in yogurt and beverages [30,35,36], to the best of the authors' knowledge, no studies have been conducted specifically investigating the effect of fermentation on hybrid cheese. In the papers listed here, only half of them used a starter culture in cheesemaking for acidification purposes. One of the studies viewed their hybrid product as a potential probiotic and recorded the evolution of the number of LAB during a two-week storage period, proving that the number did not decrease below 7.0 log CFU/g, thus confirming the probiotic potential [37]. Since fermentation with bacteria and fungi has such an importance for the traditional cheesemaking process and to improve the functionality of plant proteins [38], research conducted in this field should include these aspects as well.

Overall, these studies show that there is a low threshold for the proportion of plant bases that can replace animal milk. Indeed, most studies report a negative impact on the taste or the texture when adding more than 15%, and only a few reach values as high as 20% plant-based ingredients. Above this quantity, negative effects can be found such as collapsing of the texture, grittiness, or off-flavors. When developing products to reduce the environmental impact of cheese, such low proportions of plant ingredients will not be sufficient, and development efforts should aim for higher standards. Moreover, some studies have aimed to replace protein with starches. While the effort to include more fiber in the diet is an honorable goal, one should not forget that the main macronutrient provided by cheese is protein, and that consumers will expect alternative cheese products to be a source thereof. The formulation of hybrid cheese should therefore aim for protein contents at least as high as those of conventional cheese. Protein is a useful tool in food formulation, as it plays important roles as an emulsifier, foaming and gelling agent. However, milk and plant proteins are different in their composition and structure, making them difficult to interchange.

**Table 1.** Summary of studies investigating hybrid cheese from complex materials.

| Cheese Type | Plant-Base | Fermentation | Main Results | Reference |
|---|---|---|---|---|
| Mozzarella | Hydroxypropylated barley starches | No | Replacement of 15% of rennet casein with starches provides acceptable textures with improved meltability | Mehfooz et al., 2021 [39] |
| Mozzarella | Soy milk | Thermophilic Y 082 D (Clerici Sacco International Srl, Cadorago Como, Italy): *Lactobacillus bulgaricus, Streptococcus thermophilus* | Using 10 to 20% soy milk is acceptable. Higher proportion decreases meltability but increases nutritional profile | Jeewanthi et al., 2014 [40] |
| Feta | Lab-made lentil milk, inulin | No | Adding too much lentil protein disrupts the structure, but 10% is acceptable. Inulin increases likeability as a fat replacer | Moradi et al., 2021 [41] |
| Cream cheese | Lab-made soy protein concentrate | No | From 5 to 15 g/L soy protein concentrate added to partially skim milk; addition of this amount of soy protein did not impact sensory experience too strongly. The texture was still acceptable, taste was slightly negatively impacted. Products were stable | Rinaldoni et al., 2014 [42] |
| Cream cheese | Pea protein, lupin protein or oats protein isolates | No | All emulsions created were stable; caseins and whey proteins are primarily adsorbed at the oil/water interface | Grasberger et al., 2021 [43] |
| Cheddar | Soy milk | *Streptococcus lactis* (now *Lactococcus lactis*) | Cow's milk can be replaced with soy milk up to 15% without impairing sensory qualities (but sensory experience was made by untrained lab personal) | Rani and Verna, 1995 [44] |

**Table 1.** *Cont.*

| Cheese Type | Plant-Base | Fermentation | Main Results | Reference |
|---|---|---|---|---|
| Cheddar | Soy protein isolate (SPI) | *S. thermophilus*, *Lactobacillus delbrueckii* ssp. *bulgaricus* | Up to 7% soy protein, no adverse effect on taste, microstructure was less compact with soy protein. Advise a max 5% soy protein | Atia et al., 2004 [45] |
| Yogurt cheese | Lab-made soy milk | Thermophilic Y332A (Clerici Sacco International Srl, Cadorago Como, Italy), *L. bulgaricus* and *S. thermophilus* | Higher protein and lower fat content in cheeses supplemented with up to 20% soy milk; no significant difference in the rheology character between control and soy supplemented | Lee et al., 2015 [37] |

## 3. Achieving Texture through Raw Material Selection and Pre-Processing

Food materials, including ingredients, are complex as they can be heterogeneous, amorphous and hygroscopic while having different structural properties (e.g., morphology) [46]. The processing of food materials is often associated with modification at all levels such as micro, meso and molecular [47]. Also, the concomitant phenomena of mass transfer with thermal treatment during food processing can result in physicochemical changes. Phase transitions from liquid to solid during food processing are often encountered in systems dominant in proteins or carbohydrates due to aggregation and gelation, respectively [48]. Understanding protein physicochemical properties, modifications and interactions during processing can represent a powerful tool for modifying and tailoring physicochemical properties, appearance and texture to develop formulated new products such as plant–dairy hybrid food. The gelation of milk proteins is well understood and extensively described [49–51]. There are also extensive studies on the gelation properties of plant proteins, such as the heat-induced gelation of plant globulin [52–54], acid induction [55] or combinations of treatments [56,57]. As a flexitarian diet is becoming increasingly common among consumers due to health, environmental and sustainability concerns [58], scientific reports investigating plant–dairy systems for their physicochemical properties, as those influenced by processing conditions are emerging [59,60], with some relevant reports being summarized in Table 2.

Early studies were primarily focused on combining SPI with whey protein isolate, or micellar casein isolate, at neutral pH (i.e., 6.7 to 7.0) using mixing (or shear mixing) and heat treatment. Although these early studies used ideal conditions such as purified ingredients, there was virtually no interference of other components (e.g., lipids or carbohydrates) and standardized pH; it was fundamentally established that the addition of dairy proteins, especially whey proteins, to soy protein was modulating the rheological properties of such hybrid systems towards low viscosity, and that the heat load applied during processing influenced the formation and properties of the aggregates (soluble or insoluble).

Most recently, pea proteins and their fractions (i.e., legumin and vicilin) in combination with different sources of dairy proteins (whey, casein and a combination thereof) have been receiving a lot of attention from researchers. This draw of attention towards pea is ascribable to pea being considered safer than soy, in terms of allergenicity, regulations (GMO) and sustainability, as pea requires less water than soy to grow [61,62]. Most of the work conducted on pea protein and dairy have focused on either pea protein concentrate or isolate along with relatively simple dairy sources such as whey proteins or micellar casein proteins and skim milk [54,55,63–68]. Most of these studies processed dairy and plant proteins using simple mixing with ultra-pure water and typical standardized heat treatment processing for achieving gelation (80 to 95 °C, 20 to 60 min). In addition, some of these works also studied the influence of pH (either from acid addition or fermentation) or enzymatic treatment for providing insights relative to gelation characteristics and protein–protein interactions. At a general level, the different extents of processing (e.g., heat treatment, acidification,

etc.), plant-to-dairy protein ratio and total protein concentration significantly influenced the following, but not limited to, properties: *(a)* onset gelling temperature decreasing with a high proportion of plant protein; *(b)* plant protein produces a gel network independent of dairy protein; *(c)* rheological properties of hybrid gel (especially storage module) was influenced by the relative proportion of plant proteins; *(d)* acidification of the hybrid system was accelerated when plant proteins were dominant whilst decreasing gel stiffness and *(e)* inclusion of dairy protein increased functionality, especially when fat components were included in the hybrid system. However, there are limitations for the comparison between studies, since the technical properties of plant protein vary strongly depending on the extraction method [52], as commercial protein isolates are usually denatured during the extraction process, which lowers solubility [53,64].

Overall, one should point out that limited reports have investigated other plant protein sources, apart from pea or soy proteins, in the presence of dairy proteins. Additionally, most of the work was conducted in a protein-dominant system without including other necessary components for a fermentable hybrid cheese perspective, such as lipids and fermentable carbohydrates sourced from plant or dairy (e.g., lactose). Fat is one of the major macrochemical components of conventional full-fat dairy cheese (e.g., cheddar, Parmigiano Reggiano and gouda), which is crucial for taste, texture and flavor, with an average protein-to-fat ratio ranging from 0.70 to 1.02 [69]. It is, therefore, necessary to incorporate fat components into hybrid cheese matrices; however, only a few studies, to the authors' knowledge, can be considered fundamental from a hybrid cheese perspective, as seen above. From a plant–dairy hybrid perspective, the work of Grasberger et al. [43], carried out on fat content similar to conventional dairy cheese (i.e., 20 to 21.4%) but at a lower protein content (i.e., 8%) than regular medium-hard cheeses, is a good example. They used different plant proteins such as pea, oat and lupin in conjunction with fresh milk and whey protein in an extensive range of plant–dairy ratios (see Table 2). The formulations were processed very similarly to what is commonly carried out at the industrial level, such as the reconstitution of the ingredients and pre-homogenization, followed by homogenization at 250 bar, with a thermal treatment (85 °C for 5 min) to generate protein gelation. Interestingly, dairy protein inclusion was crucial to stabilizing the oil droplet at the interface, while plant protein modulated the gel's final characteristics, with lupin and pea resulting in creamy spreadable-like cheese in contrast to oat protein, which resulted in the gel having pasta-filata-like properties. Similarly, a very interesting work by Canon et al. [36] focused on hybrid set-type yogurts fermented using LAB strains (*Lactiplantibacillus plantarum* and *Enterococcus faecalis*). The yogurts' formulation was relatively higher in protein content (6.6%) than their conventional dairy counterparts. The proteins were supplied from lupin protein isolate and whey protein or skim milk powder as dairy ingredients along with the inclusion of milk and plant fat components (milk fat solids and coconut at 1.5%). It was shown that pre-treatment such as homogenization (two steps, total 300 bar) and pre-heating (95 °C for 10 min) prior to fermentation of the protein suspension can improve the texture with a higher content of lupin, further modulating toward a high apparent viscosity. Pre-treatment of the protein ingredient solution prior to the mixing stage produces hybrid mixtures with different textures and gel structures than those not being pre-heat treated, a finding established also in other published reports [70–72]. However, a milk/lupin protein ratio of 67:33 was more acceptable compared to the 50:50 ones, as coconut oil did not negatively impact the overall sensory properties. Fermentation also improved acceptability by producing more and different aroma compounds and a more pleasant texture due to acidity levels close to the isoelectric point of dairy–plant proteins.

Although there is an incremental number of studies on understanding the overall properties of plant–dairy hybrid systems in terms of physicochemical properties and structure whilst correlating these properties to processing and pre-processing, there is limited evidence on mimicking or reproducing plant–dairy hybrid systems that can resemble conventional dairy products. Most of these works were carried out in model conditions (composed only of protein-based ingredients) without being emulsified (no fat included).

However, the advantage of combining the pre-processing of the plant and dairy ingredients alone or in combination thereof, followed by fermentation using a combination of dairy and plant carbohydrate, appears promising for improving and making plant-dairy hybrid systems close to dairy. Although more reports in this area are necessary, this may represent a very significant advancement toward developing plant–dairy hybrid products (e.g., cheeses and spreadable creams) for sustainable diets aiming for a gradual and full transition towards plant-based diets.

**Table 2.** The relevant literature on main effects and properties of different plant–dairy hybrid systems.

| Ingredients and Concentrations | Plant to Dairy Ratio | Processing conditions | Highlights | Reference |
|---|---|---|---|---|
| SPI, WPI—6% (*w/v*) protein content | 0:100, 30:70, 50:50, 70:30, 100:0 | Mixed; 90 °C 60 min; pH 7.0 | While soy protein in isolation formed large, soluble aggregates, the addition to whey protein significantly reduced the amount of soluble aggregate and induced precipitation. 7S and the basic subunit of 11S were present in the precipitate after heating with WPI. | Roesch and Corredig (2005) [73] |
| SPI, MCI—10 and 15% (*w/w*) protein content | 50:50 | High shear mixing; 40, 60 and 95 °C 15 min; native pH (6.74–6.86) | Temperature above the denaturation of soy glycinin induced aggregation and gelling via disulfide bonding in both soy and casein; below soy protein critical gelling concentration, mixtures resulted in a Newtonian liquid with lower viscosity and improved storage stability compared to non-denaturing heat treatment. | Cosmin and Moraru (2013) [74] |
| PPC, WPC—10, 16 and 22% (*w/v*) protein content | 80:20, 50:50, 20:80 | Mixed, 92 °C 30 min; pH 4.0, 6.0, 8.0 | Higher synergistic enhancement at 20:80 pea/whey was observed for gels at pH 6.0; 50:50 and 20:80 ratios both had synergistic viscosity enhancement after heat treatment. | Wong et al. (2013) [63] |
| PPC (legumin and vicilin fractions), Purified casein micelles—1.8% (*w/v*) protein content | 50:50 | Mixed; 85 °C 60 min; pH 7.10 | Casein micelles stabilized pea proteins against heat-induced unfolding; vicilin, in presence of casein micelles, produced heat-induced soluble aggregates, while legumin produced insoluble aggregates. | Message et al. (2017) [54] |
| PPI, SMP—14.8% (*w/w*) protein content | 50:50 | Mixed, (a) heat treated 90 °C 60 min; (b) acidified with glucono delta-lactone 2% (*w/v*) and (c) Enzymatic treatment CaSO4 (0.3%), chymosin (0.5%), and TGase (0.3%) | Pea gels induced by acid or enzyme had a higher storage modulus (G0). Thermal treatment induced covalent bonds. Enzyme-induced gels produced coarse aggregates with a more excellent resistance to strain. | Ben-Harb et al. (2018) [64] |
| MPC, WPI, PPI, SPI—12% (*w/w*) Protein content | 50:50, 58:41, 66:33, 75:25, 83:16, 90:10, 100:0 | Stir mixed; 90 °C 1 h; pH 7.00 | Gelation temperature increased with plant protein, with soy being more effective than pea protein. | Silva et al. (2018) [65] |
| PPI (globulin fraction); WPI (β-lactoglobulin fraction)—2% (*w/v*) Protein content | 0:100, 30:70, 50:50,70:30, 100:0 | Stir mixed; 85 °C 60 min; pH 7.20 | Synergistic effect for increased gels elasticity and water holding capacity compared to gels containing pure aggregates of Glob or mixtures of Glob and βlg aggregates. | Mohamed–Lazhar Chihia et al. (2018) [66] |

**Table 2.** *Cont.*

| Ingredients and Concentrations | Plant to Dairy Ratio | Processing conditions | Highlights | Reference |
|---|---|---|---|---|
| RPI, WPI—0.0 to 20% (*w/w*) Protein content; | 90:10, 80:20, 70:30, 60:40, 50:50, 40:60, 30:70, 20:80, 10:90 | Mixed, 80 to 95 °C 30 min; pH 7.00 | Independent gel network formation: higher plant to dairy ratio produced a stronger gel than single standalone ingredient. | Ainis et al. (2019) [75] |
| OPC, OPI, SMP—12.3, 13.8 and 15.3% (*w/w*) total solids | 60:40 | Homogenized one-pass 200 bar; 80 °C 20 min; native pH | OPC good replacer of SMP; good functionality due to oat starch. | Brückner-Gühmann et al. (2019) [76] |
| PPI, OPC, LPI, WPI, SM—8.0% (*w/v*) Protein content; 20 to 21.4% (*w/v*) fat content | 33:66, 32:67, 26:73, 67:32, 67:32, 68:31, 70:30, | Mixed, homogenized one-step 250 bar; 85 °C 5 min; pH 6.1 to 6.6 | Emulsion gel oil droplets were stabilized by dairy proteins; LPI and PPI induced low onset gelation temperature. | Grasberger et al. (2021) [43] |
| LPI, SMP, WPI, milk fat and coconut oil—6.6% (*w/w*) Protein content; 1.5% (*w/w*) fat content | 50:50, 67:33 | Stirred and homogenized two-step 250/50 bar; heated 95 °C 10 min; pH native; LAB fermented | Fermentation improved the texture and reduced off-flavor of lupin-dominant formulation. | Canon etl al. (2022) [36] |
| SPI, WPI—4% (*w/v*) protein content | 100:0, 75:25, 50:50, 25:75, 0:100 | Mixed; 95 °C 30 min; pH 7.00 then acidified | SPI inclusion decreased stiffness (low G′) and stretchability (lower γc) of acid-induced gels; hybrid gels displayed a relatively more elastic response in the nonlinear viscoelastic regime with a plastic behavior. | Xia et al. (2022) [67] |
| PPI; SMP—5.0, 7.0, 9.0, and 11% (*w/w*) Protein content | 27:73, 33:66, 42:57, 60:40 | Stir mixed; 85 °C 5 min; pH 6.30 to 6.80 | The presence of pea proteins accelerates acid-induced gelation but weakens the structure of mixed gels. | Oliveira et al. (2022) [68] |

**Dairy ingredients:** Whey protein concentrate (WPC); whey protein isolate (WPI); fresh skim milk (SM); skim milk powder (SMP); milk protein concentrate (MPC) and micellar casein isolate (MCI). **Plant ingredients:** Soy protein isolate (SPI); pea protein concentrate (PPC); pea protein isolate (PPI); oat protein concentrate (OPC); oat protein isolate (OPI); lupin protein isolate (LPI) and rapeseed protein isolate (RPI). 4. Flavors, off-flavors and antinutritional factors in hybrid cheese.

For the development of hybrid cheeses, the combination of plant-based ingredients with milk may lead to the generation of nutritional, technological, and sensorial challenges when aiming to mimic their dairy-based counterparts [13,77]. Implementing new raw materials will inevitably demand the application of new methods to reach a convincing flavor profile. In a fully milk-based cheese production, many parameters already affect the flavor profile of the resulting cheese. Animal origin [78], diet [79], physical treatment (i.e., pasteurization, cooking temperature and salting) [80] and ripening conditions (i.e., microbiota, storage temperature and maturation time) [81] have all been thoroughly studied since they have a significant effect on the final product. The introduction of plant-based ingredients will equally be influenced by these steps, and one will need to carefully study how to maintain favored properties, while avoiding unwanted inputs like off-flavors and antinutritional factors (Figure 1).

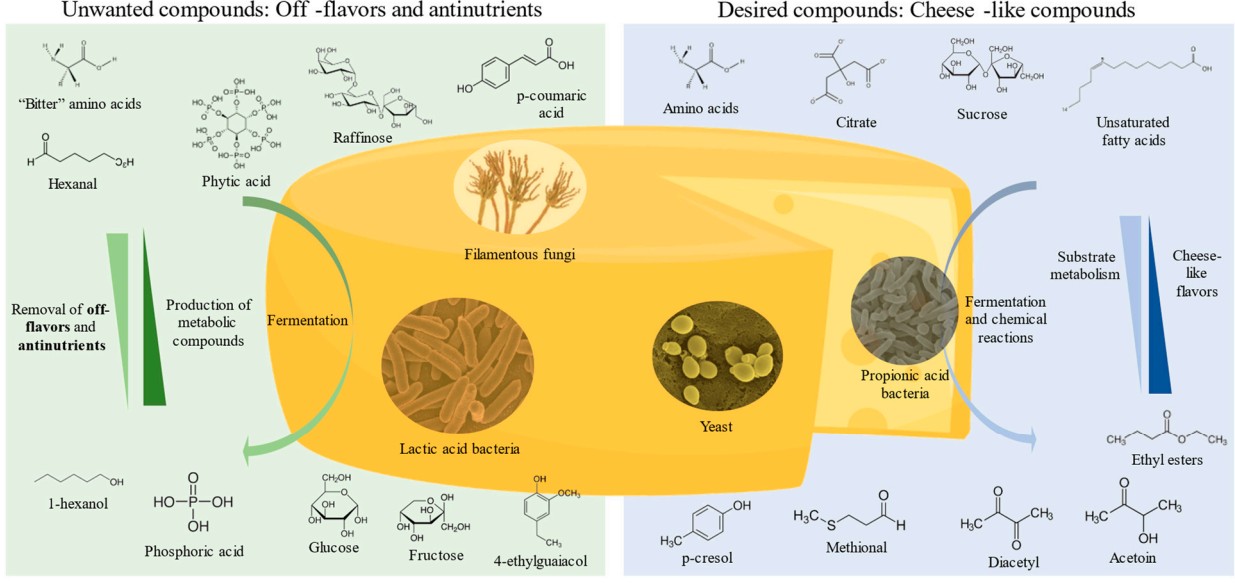

**Figure 1.** Metabolic pathways of the production and utilization of desired or undesired compounds by LAB, propionic acid bacteria, yeast and molds in the context of plant and milk fermentation.

Plant-based raw materials are characterized by containing antinutritional factors such as phytates, enzymatic inhibitors and saponins that affect the absorption and bioavailability of valuable compounds [9,77,82]. Phytates reduce the mineral bioavailability of metal ions by forming complexes, while enzymatic inhibitors, such as trypsin inhibitors present in legumes, decrease plant protein digestibility [82,83]. Furthermore, saponins affect the absorption of specific vitamins due to their cross-interaction with fat-soluble vitamins that are chemically similar to plant sterols [82]. Different antinutrients are found depending on the plant-based raw material used. For instance, cereals and legumes contain flatulence-causing oligosaccharides such as raffinose and stachyose.

Off-flavor molecules are classified into volatiles, such as aldehydes, alcohols and ketones, and non-volatiles such as phenolic compounds, peptides and saponins [84]. The combination of ingredients from animal and plant origin in hybrid cheeses can result in off-flavor notes from both compounds, influencing the products' sensorial properties and consumer acceptance [84,85]. First, milk itself can provide off-flavor compounds, which could originate from physical (milk pasteurization and light exposure)-, enzymatic (oxidation and lipolization)-, non-enzymatic (Maillard reaction)- and microbial (protein, amino acid, carbohydrate and lipid metabolisms)-driven reactions, which could impart "sulfurous", "bitter" and "rancid" notes [86]. However, plant-based dairy alternatives are known for their typical off-flavors, often characterized as "green" or "beany", and are therefore a significant source of undesired compounds [77,84,87]. The desirability of cheese flavors strongly depends on the type of cheese and can even be considered a defect when in the wrong cheese. Therefore, specific starter cultures and processing conditions are optimized for each type [88]. For instance, "sulfurous" notes produced from sulfur amino acids such as methionine are desired in camembert and cheddar cheeses [88], while undesired in parmesan cheese. Nevertheless, the bitterness caused by peptides containing hydrophobic amino acids is usually considered "unwanted" in most types of cheeses [88].

Volatile off-flavors that impart "green" notes are mainly caused by the degradation of polyunsaturated fatty acids through oxidation reactions, enzymatically and non-enzymatically driven [84]. Enzymatically, they are mainly started by the action of lipooxygenases (LOX) that degrade plant fatty acids such as linoleic acids and transform them into hydroperoxide intermediates, producing fatty acid aldehydes such as hexanal, pen-

tanal and nonanal [84,85,89]. Those are converted into their respective alcohol and ketone compounds via enzymatic and non-enzymatic reactions, respectively [84]. Non-volatile off-flavors are characterized by giving off astringent and bitter taste notes [9]. Phenolic compounds such as ferulic, coumaric and gallic acids, saponins, alkaloids, peptides and amino acids are some of the examples [9]. Also, it has been demonstrated that specific combinations of volatile off-flavor molecules could enhance (or reduce) their perception by synergistic (or antagonistic) effects with other volatiles [90].

Different approaches have been investigated to reduce the concentration of off-flavors and antinutrients when developing plant-based fermented dairy alternatives [87]. Physical-based technologies such as roasting, dehulling, soaking and blanching and milling have been implemented in sesame, legumes, soy and peas and soy, respectively [87]. For instance, roasting, milling and soaking have been applied to reduce LOX activity, reducing lipid oxidation and off-flavor formation [91–93], while the dehulling of legumes has been used to decrease phytate concentration [91]. Chemical-based technologies such as pH alterations (alkalinization), chemical deodorization in pulses (pea, lentil and soy) and pseudocereals (quinoa) have been applied for texture purposes as well as for reducing LOX activity and off-flavor generation [87]. The pasteurization process of milk, preceding the initiation of the regular cheesemaking process, also alters the flavor profile, depending on the degree and type of pasteurization [94,95].

Finally, fermentation-based approaches have been investigated due to the fermentation step necessary to produce dairy and hybrid cheeses [87]. In the following, we will elaborate on the contribution from fermentation-based solutions in terms of flavor alterations, as it is the most significant contributor in cheesemaking.

In traditional cheesemaking, large parts of the flavor formation happen during the ripening stage, where the secondary metabolism of the present bacteria, yeasts and molds create the main part of flavor molecules associated with cheese [96]. Understanding the individual role, as well as the interplay of these organisms, is essential if fermentation is applied in the production of a hybrid product. Since these products will have a new substrate baseline, it will be important to tailor the use of specific strains to secure satisfying alternatives. Specific LAB strains are extensively used in the production of dairy cheeses as starter cultures, and are responsible for generating texture, flavor and other desired properties in different types of cheeses [97]. In hybrid cheese production, the plant-based substrate added will affect the metabolism and growth performance of the starter cultures added, as well as the organoleptic properties of the final product. Therefore, dairy-adapted microorganisms might not be optimal when aiming to alter the flavor profile and remove antinutrients present in hybrid cheeses [98]. LAB strains with specific gene clusters and enzymatic activities should be selected and added for that purpose [98–100]. As an example, the high activity of LAB alcohol and aldehyde dehydrogenases has been correlated with the removal of aldehyde molecules such as hexanal in plant-based dairy alternatives [89]. Also, high activities of β-glucosidase, α-rhamnosidase and β-galactosidase have been related to the degradation of the sugar moieties attached to saponins and glycoalkaloids [101–103]. Depending on the plant-based substrate, different sugars are combined to their respective saponins [104]. For example, in soyasaponins, avenacosides (A and B), α-solanine and α-chaconine, are bonded with dimers and trimers of glucose, galactose, arabinose and/or rhamnose through different O-glycosidic bonds to each other and to their respective saponins [104,105].

In a dairy cheese context, the flavor contribution from fermentation can be split into three categories of origin: protein-, lipid-, and carbohydrate-derived flavor compounds [96]. The carbohydrate metabolism is almost exclusively linked to primary metabolism, generating compounds such as lactate, acetate, ethanol, $CO_2$ and the dairy-related diacetyl, acetoin and butanediol [90], usually linked to fermented dairy products. From lipid catabolism (lypolysis), the initial breakdown of triglycerides provides free fatty acids (FFA), the precursors for flavor compounds such as lactones, esters, alcohols and methylketones [106,107]. Many of these impart fruity and herbal flavors and are rarely linked to LAB fermentation,

although there are studies showing the significant release of FFA from LAB [108]. Lipolysis is instead often linked to *Propionibacterium* [109] and mold-ripened cheeses [107]. Flavors related to the breakdown of proteins (proteolysis) are produced by both the starter culture strains as well as commensal strains of bacteria, yeasts and molds [110–112]. However, proteolysis is initiated by the addition of rennet and the activity of cell-envelope proteinases and peptidases from the starter LAB [112]. The resulting release of small peptides and free amino acids imparts a savory and salty flavor but is most importantly fed into the amino acid catabolism, which produces sulfurous aromas and keto-acids [111]. The proteolytic nature of many LAB is possibly linked to them being auxothropic in their amino acid anabolism, hence needing to collect certain amino acids from their surroundings [113]. However, as most dairy-adapted LAB strains have been selected for their ability to break down the caseins in milk, little focus has been drawn on finding strains suitable for the proteolysis of plant proteins. Studies have been conducted on the extracellular microbial proteases, specifically targeting plant proteins [38]. Even though there are some LAB strains showing proteolytic effects on certain plant proteins, molds and yeasts are far more capable [38]. Molds are essential for the ripening of certain cheeses such as camembert and Roquefort, and many mold-ripened PBCAs have appeared on the market. However, the safety of such molds should be carefully assessed when changing the growth substrate from milk to plant bases, since it could induce the production of mycotoxins. This topic is rarely addressed and should receive more attention as the development of fungi-fermented foods is gaining momentum [114].

To conclude the above, metabolic capabilities of the starter cultures added to the fermentation of hybrid cheeses, including LAB, yeasts and non-LAB strains such as *Propionibacterium freudenreichii*, need to be investigated regarding the production of cheese-related flavors, and the elimination of antinutrients and off-flavors in a hybrid cheese products. Metabolic pathways and their actual application in this field are currently being studied for each of the off-flavor and antinutrient molecules for the overall organoleptic and functional improvement in hybrid cheese production.

## 4. Concluding Remarks and Perspectives

The food system plays a prominent role in the challenges that society is facing in terms of climate change and its impacts on food security. These are clear warning signs that the food industry has an urgent need to adapt to more sustainable food production practices. The increase in market demand for PBA is an incentive towards increased product development. While plant-based dairy alternatives have not yet been developed to a satisfactory point in terms of taste and nutrition, this review shows the great potential for developing hybrid cheese products. Key challenges in this domain are the interaction of plant and milk protein as well as sensory properties. Since cheese is a key source of protein in the diet, a hybrid product should have comparable or superior nutritional, textural and sensorial profile. Therefore, research should aim to find processes to make plant proteins more functional, so they can provide unique functionalities in the hybrid cheese matrix. Furthermore, focus should be drawn to fermenting such products, as fermentation represents a great opportunity to reduce the off-flavor characteristics for plant-based ingredients, while adding desired cheese aromas and nutritional compounds such as vitamins. More systematic investigations are needed, both in terms of protein interactions to provide target textures and sensory analysis to satisfy consumer preferences and succeed in a fast transition.

**Author Contributions:** B.M.L.G. and C.H.B.-B. conceptualized the review framework. All authors contributed to the design of the review. B.M.L.G. made the first draft of the manuscript with large contributions from A.P.W., G.B. and G.E.S.M. Figures and tables were made by A.P.W., G.B., G.E.S.M. and B.M.L.G. The draft was reviewed by L.M.A., K.A., E.B.H. and C.H.B.-B. All authors have read and agreed to the published version of the manuscript.

**Funding:** This review was supported by a Danish Dairy Research Foundation grant (SusCheese). B.M.L.G. and G.B., G.E.S.M., E.B.H., L.M.A. and C.H.B.-B. were supported by an Innovation Fund Denmark grant—Innomission AgriFoodTure called REPLANTED. A.P.W. was funded by a joint PhD alliance grant between Technical University of Denmark (DTU) and University of Queensland (UQ). The funding source was not involved in the design of the review, writing of the review or the decision to submit the review for publication.

**Conflicts of Interest:** The authors from DTU and UCPH affirm that the review was conducted without any commercial or financial relationships that could be interpreted as possible conflicts of interest. K.A. is an employee of the Danish dairy Thise.

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
