# Peer review of "Hybrid Cheeses—Supplementation of Cheese with Plant-Based Ingredients for a Tasty, Nutritious and Sustainable Food Transition"

_fermentation, doi:10.3390/fermentation9070667_

Round 1
Reviewer 1 Report
The manuscript entitled "Hybrid-cheeses – supplementation of cheese with plant-based ingredients for a tasty, nutritious and sustainable food transition" is well written, organized and well structured paper on an actual and relevant topic. It covers all important issues regarding plant based cheese alternatives, including not only sensory properties and nutritional value, but also their environmental impact. In my opinion the manuscript will be very attractive for readers and industries and especially useful for researchers from the field, since it highlights the drawbacks of currently available products and discuss limiting parameters of previous studies, giving clear directions for the future ones. For example, the authors suggest special focus on investigating the effect of fermentation on hybrid cheese. They also pointed out the importance of including more plant protein sources in the future investigation as well as other necessary components for a fermentable hybrid cheese perspective like lipids and fermentable carbohydrates. I recommend this manuscript to be published. The only suggestion to authors is to check figure designation (there is only one figure, mentioned in text as Fig 1, but designated as Fig 2).
Author Response
Thank you for taking the time to review our manuscript, we appreciate your feedback and positive comments.
Reviewer 2 Report
The topic of the manuscript: „Hybrid-cheeses – supplementation of cheese with plant-based ingredients for a tasty, nutritious and sustainable food transition” falls within the thematic scope of FERMENTATION.
The manuscript discusses issues related to the current state of knowledge about the possibility of supplementing cheese with plant ingredients to ensure a tasty, nutritious and balanced nutritional transition. Thank you very much for the opportunity to review this article. I read the manuscript with great interest. It is prepared in a very appropriate way and stimulates the reader to their own reflections. I found only a few minor errors. I also have two suggestions: one for Figure 2, and the other (Chapter 3, paragraphs 454-460 or Chapter 4) for Authors' consideration only.
All comments were introduced in the review mode to the attached pdf file.

Author Response
Thank you for taking the time to review our manuscript and your thorough comments. All the formatting issues have been addressed.
The comment about the use of molds in ripening has been addressed in a few lines (L475 – L481). This is a subject that will need more attention in further research, as to the best of our knowledge, no literature exists on the comprehensive safety assessment of mold ripened plant-based cheese.
Reviewer 3 Report
The paper is very well written. The following suggestions may help authors to further improve the manuscript:
L94 Dairy products, and particularly cheese, are important sources of macronutrients (protein and fats) and micronutrients (calcium, vitamin B2, vitamin B12, vitamin D and iodine) in the diet [22,23]. Switching from animal milk to PB matrices creates a significant difference in nutritional intake which can lead to severe deficiencies of which consumers are not always aware of… Supplementation in micronutrients has proven to be an efficient way to achieve nutritional adequacy, and efforts should be made in formulation to reach an acceptable protein and healthy fat content.
Iron fortified foods and iodized salt are often found on the market. Can the authors write something more about the possibility of supplementing plant-based and hybrid cheeses with additional calcium and iodine. Also, is the fortification of hybrid cheeses with various nutrients sufficient as a substitute? Is there any study that deals with supplementation of plant-based and hybrid cheeses with fruit and vegetable pomaces, which often represent a reusable source of nutrients? Bearing in mind that in this paper it is emphasized that the transition from conventional cheeses to hybrid cheeses is important for the environment, perhaps the authors could discuss something about cheeses with fruit and vegetable pomace.
Soy protein isolate abbreviation (SPI) is introduced first time in Table 2 L336, but appears earlier in the text in L231 Table1 and L251 Early studies were primarily focused on combining soy protein isolate
Please introduce abbreviation when appears first time in the text.
L462 Instead Table 2 should stand Table 1.
Author Response
Thank you for your feedback, we appreciate the time you took to review our manuscript.
While we wanted to englobe all aspects of cheeses and their alternatives as food products, nutrition is on the edge of the scope of this review. However, the paragraph treating this aspect (L90 to L106) has been rewritten for more clarity and details. Fortification with calcium and iodine is usually done in liquid or semi-solid (yogurt) products but rarely on solid (cheese) products. It can be however achieved with different ratios of soluble and insoluble salts to avoid disruption of protein structure or sedimentation. While this is not the focus of this review, the topic should be addressed in fundamental research for further investigations.
The use of pomaces as material in the cheese matrix is a very interesting point. It is without doubt that the use of side streams is necessary to increase the sustainability of the food system. To our knowledge, the field of hybrid cheese is in very early stages, where the focus is for now mainly on understanding the dynamic interactions between plant and milk protein. Investigations are therefore conducted on “easy” material rather than on complex and fluctuating sources such as pomaces. The suggestion has been addressed in a few lines (L132 – L137).